# Improving the X-ray Shielding Performance of Tungsten Thin-Film Plates Manufactured Using the Rolling Technology

**Seon-Chil Kim** 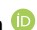

Department of Biomedical Engineering, School of Medicine, Keimyung University, 1095 Dalgubeol-daero, Daegu 42601, Korea; chil@kmu.ac.kr; Tel.: +82-10-4803-7773

**Abstract:** X-ray shields used for medical purposes are manufactured using lead, which is inexpensive and easy to manufacture. However, as lead can be a major factor contributing toward environmental contamination, such as lead poisoning, a radiation-shielding plate was manufactured in this study using a tungsten plate, an eco-friendly material, through a rolling process at different temperatures. In addition, the shielding plate produced via the hot-rolling method exhibited better shielding performance than that of the plate produced using the cold-rolling method, and the multilayer structure was well formed, as indicated in the cross-sectional image analysis. Upon applying a peak voltage of 100 kVp to the X-ray tube, the shielding performance observed was 80% and 96% when the plate thickness was 0.1 mm and 0.3 mm, respectively. Therefore, it is expected that, in the future, the pure tungsten-based shield presented in this study will replace lead plates, owing to its superior standardization and reproducibility of the shielding performance.

**Keywords:** lead; tungsten; radiation shielding; medical radiation; shielding plate

## 1. Introduction

Lead is the most widely used material for radiation shielding in medical institutions [1]. Because of its excellent processability, it is sometimes used as a plate or mixed with a polymer material to produce a sheet or film [2]. However, lead is classified as a heavy metal; thus, the environmental concerns owing to its toxicity have led to a reduction in its use [3]. Therefore, an eco-friendly material with excellent shielding performance, processability, and compatibility with other materials is a topic of research interest [4,5]. Tungsten, a substitute for lead, is one of the most widely used radiation-shielding materials in medical institutions [6]. It has an atomic number of 74, an atomic weight of 183.84 g/mol, and a density of 19.25 g/cm$^3$, with a shielding performance comparable to that of lead [7]. Tungsten is mainly used as tungsten oxide or tungsten carbide, and it is difficult to manufacture a tungsten shield in a desired shape by directly processing it due to its high melting point of 3400 °C [8,9]. However, as it shows excellent affinity with other composite materials, tungsten can be used as a shield of various types when mixed suitably [10].

Typically, X-ray shields used in medical institutions are made of thin plates by mixing lead, tin, and copper [11]. However, challenges with disposal and the weight of the product restrict its application [12,13]. Therefore, a lighter and eco-friendly shielding plate with an appropriate thickness and size would be more convenient to protect patients and medical staff in medical institutions.

The metallic shielding plates primarily used for radiation shielding in such institutions can be used as shields during X-ray imaging and shielding parts inside medical devices, and for radiation-shielding carts used to transport radionuclides and nucleation injection cylinders used in nuclear medicine [14].

A processing method that retains strength and shielding performance is essential for the manufacturing of such products. For a tungsten-based shielding plate, it is necessary

to establish a standard for the shielding performance according to thickness, and the most important aspect in the manufacturing process is the reproducibility of the shielding performance [15]. The shielding performance and lead equivalence of a previous lead shield were evaluated based on its thickness [16]. Therefore, it would also be possible to determine the shielding performance of the tungsten equivalent (mmW) with respect to its thickness, similar to that of an aluminum (mmAl) or lead equivalent (mmPb).

In recent times, powder metallurgy has been widely applied in the production of tungsten shields with specific shapes [17,18]. Powder metallurgy is a method wherein particles are sintered using heat at 1800 °C or higher. Because the powder purity and particle size of tungsten directly affect its density, the processing design of the initial particles affects the shielding performance [19–21]. In addition, new tungsten machining methods must be considered for application in small medical-device-shielding components, nuclear syringe cylinders, and shielding linings of special shielded suits [22]. In this study, the rolling method, which is a tungsten-plate-based thin-film processing method, was applied instead of conventional powder metallurgy to manufacture radiation-shielding plates used in hospitals [23].

Two types of rolling methods are used for manufacturing pure tungsten plates: cold and hot rolling. The former is performed at 800–1000 °C, and the latter ranges from 1550 °C to 1900 °C or higher [24,25]. In the rolling process, pre-annealing is utilized only during hot rolling, and forging is performed up to a predetermined thickness for a certain duration [26]. In this study, an experiment was performed with a tungsten plate of the same thickness to find the difference in shielding performance depending on the applied rolling method for the plate manufacturing. In addition, it suggests a commercialization direction for manufacturing a tungsten shielding plate that can be evaluated by thickness like the existing lead plate.

## 2. Materials and Methods

During radiation shielding, energy is dissipated through interaction with the elements in the medium as the radiation passes through, with an intensity that is directly proportional to the thickness of the shielding material [27]. The degree of dissipation of the radiation energy is called the linear attenuation coefficient of the medium [28]. We assume that the initial intensity of the radiation is $I_0$, and the linear attenuation coefficient is $\mu$. When direct radiation passes through the medium, the passed intensity ($I$) when penetrating through its thickness, $t$, is expressed in Equation (1) [29] as follows:

$$I = I_0 e^{-\mu t} \tag{1}$$

where $\mu$ varies with the energy intensity of the radiation, and $t$ is expressed as the thickness of the material through which the radiation, such as the mass or number of atoms per unit area, passes. As a result, the shielding performance of the plate can be determined using the thickness of the shield and the linear attenuation coefficient. Therefore, during manufacturing, a method to increase the density is frequently utilized to increase the mass per unit area.

When a shield is manufactured by applying the rolling method to tungsten, which is commonly used as an eco-friendly shielding material, the density can be fundamentally higher than that obtained when using the conventional method of mixing tungsten powder with a polymer resin. However, a suitable processing technology that reduces thickness and increases processability is required for the manufacturing process.

Firstly, to compare and evaluate the shielding performance of the tungsten plate, changes in the shielding performance based on the processing technology and thickness were analyzed. The average particle size of the tungsten powder for the plate used in this experiment was 4.28 + 0.1 μm, and the purity was over 99%.

Using a 1 mm plate (Beijing Tianlong tungsten Technology Co. Ltd., Beijing, China), 0.3 mm and 0.1 mm plates were manufactured via hot rolling, and 0.1 mm plates were

manufactured via cold rolling. The composition of the manufactured tungsten plate was analyzed via energy-dispersive X-ray spectroscopy (EDS, Hitachi, HD-2300, Japan).

Hot rolling was performed at 1550 °C, which is 0.22 times the melting point of tungsten, and the roll speed for both the upper and lower rolls was 80 mm/s. Cold rolling was performed at a relatively low temperature of 850 °C. In addition, the cumulative reduction during the hot-rolling process was adjusted based on the 0.3 mm and 0.1 mm thicknesses, and the thickness was adjusted via forging. The rolling process can increase the density while removing internal porosity through the application of pressure to the solid-state material [30]. Moreover, the mechanical strength can be increased by constructing a dense particle structure. Although cold and hot rolling entail similar manufacturing processes, the heat treatment process applied to the tungsten plate differs with temperature. The tungsten plate manufactured via hot and cold rolling was observed using an optical microscope (FESEM; field-emission scanning electron microscope, Hitachi, S-4800). The density of the internal structure was visually compared and analyzed by observing the surface and cross-section at the same magnification.

In addition, to evaluate the shielding performance of the tungsten plate, a diagnostic X-ray generator (Toshiba E7239, 150 kV-500 mA, 1999, Tokyo, Japan) was used. As is the case in medical radiation, the tube current was fixed at 200 mA, and the experiment was conducted at tube voltages of 40 kVp, 60 kVp, 80 kVp, 100 kVp, and 120 kVp, consistent with the range used in human radiography. A radiation dose detector, Mo.9517 Radiation Monitor (Radcal Corporation, Monrovia, CA, USA), and an Mo10×5-6, 6 cc ion chamber (Radcal Corp.) were used. Both radiation detectors were inspected and calibrated before use. Figure 1 shows the experimental setup for the evaluation of the shielding performance of the tungsten plate manufactured via the rolling method. The equivalent test method for the X-ray protection of products used in Korean Industrial Standards (KS A 4025: 1990, 2009 confirmation) was applied [31].

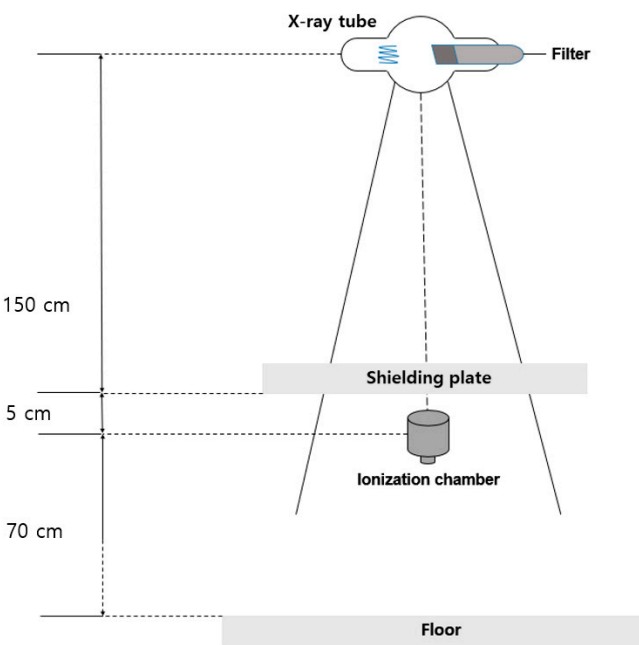

**Figure 1.** Schematic of the radiation-shielding measurement.

The shielding rate of the tungsten plate was calculated as the ratio of the irradiation dose to the transmitted dose through a plate manufactured with a certain thickness, as shown in Equation (2) [32]:

$$\text{SR}(\%) = \left[ \frac{ID_W - TD_W}{ID_W} \right] \times 100 \tag{2}$$

where SR is the shielding rate (%), $TD_W$ is the transmission dose, and $ID_W$ is the irradiation dose.

The shielding performance of three tungsten plates was evaluated for each tube voltage. The average value was calculated by irradiating each plate ten times, and the shielding performance of the tungsten plate was compared based on the thickness and processing method for each voltage. These values were then compared with the shielding performance of a lead plate with the same thickness.

## 3. Results

Three types of shielding plates were manufactured using a tungsten plate, as shown in Figure 2. Although there was no difference in appearance, the difference in thickness could be distinguished through measurements.

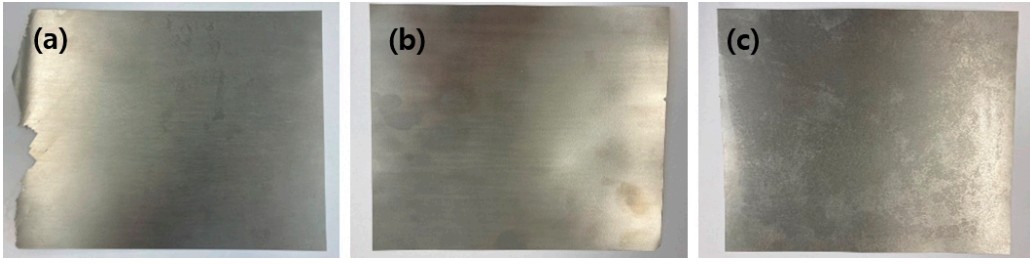

**Figure 2.** Tungsten plates. (**a**) A $0.3 \pm 0.002$ mm ($42.5$ g/cm$^3$) tungsten plate manufactured via hot rolling; (**b**) a $0.1 \pm 0.001$ mm ($21.2$ g/cm$^3$) tungsten plate manufactured via hot rolling; and (**c**) a $0.1 \pm 0.001$ mm ($19.1$ g/cm$^3$) plate manufactured via cold rolling.

Figure 3 shows the EDS component analysis of the tungsten plate fabricated in this study. The plate contained tungsten with a purity of 99% or above.

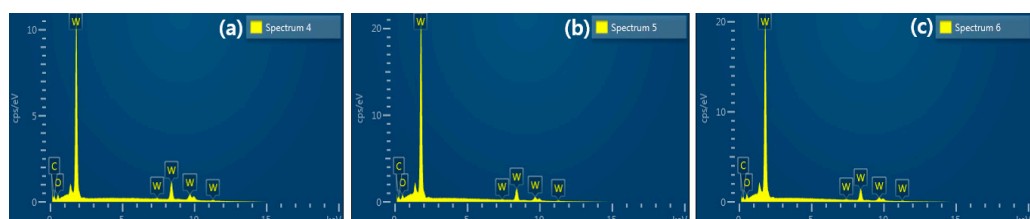

**Figure 3.** Energy-dispersive X-ray spectroscopy component analysis of tungsten plate. (**a**) Hot-rolling process for the 0.3 mm plate, (**b**) hot-rolling process for the 0.1 mm plate, (**c**) cold-rolling process for the 0.1 mm plate.

Figure 4 shows the results of the cross-section analysis of the same 0.1 mm-thick tungsten plate using an optical microscope. Figure 4a,b present plates manufactured by hot and cold rolling methods, respectively. No difference was observed in the cross-section during the two processes. However, a comparison between Figure 4(a-1,b-1) reveals a difference in the height between the layers indicated by the yellow dashes, which appeared in the form of a multilayer structure of a thin film during the hot-rolling process. Despite the use of similar multi-layer structures, the structure after hot rolling appeared to be irregular, while the structure after cold rolling appeared to be regular.

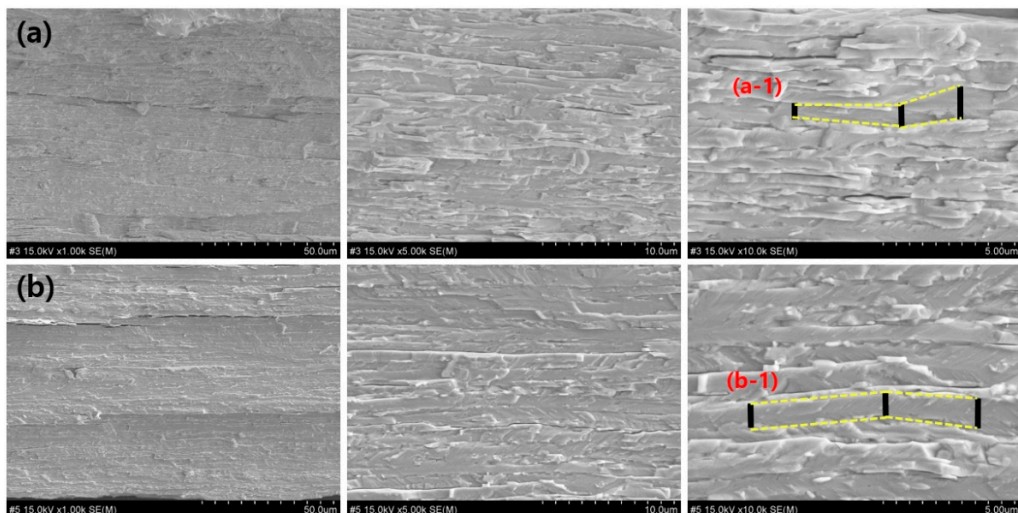

**Figure 4.** Sectional analysis of tungsten plate: (**a**) hot-rolling method; (**b**) cold-rolling method. (**a-1**) shows an irregular laminated structure, and (**b-1**) shows a regular laminated structure.

Changes to the surface of the tungsten plate due to the different rolling methods were observed, as shown in Figure 5, and it was found that the surface was slightly denser in the 0.1 mm (21.2 g/cm$^3$) plate that was repeatedly forged to reduce the thickness. The process of increasing the density or adjusting the thickness can directly affect the shielding performance.

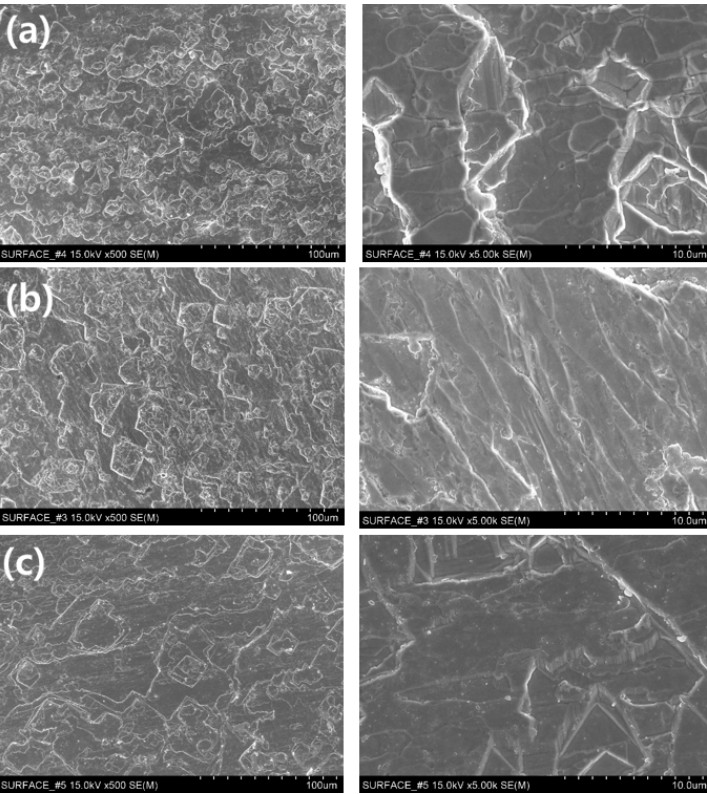

**Figure 5.** Surface analysis of tungsten shielding plate. (**a**) A 0.1 mm tungsten plate using hot rolling; (**b**) a 0.3 mm tungsten plate using hot rolling; (**c**) a 0.1 mm tungsten plate using cold rolling.

Table 1 presents the shielding performance of the tungsten shielding plate. As tungsten was used, all the manufactured plates generally showed high shielding performances. The

shielding performance of the hot-rolled plates was superior to that of the cold-rolled plate. The 0.3 mm plate exhibited the best shielding performance.

**Table 1.** Shielding performance according to the shielding plate manufacturing process.

| Effective X-ray Energy (keV) | Peak Voltage (kVp) of the X-ray Tube | Mean of Exposure (μR) | | | | Shielding Rate (%) | | |
|---|---|---|---|---|---|---|---|---|
| | | Nothing | Hot Rolling | | Cold Rolling | Hot Rolling | | Cold Rolling |
| | | | 0.3 T | 0.1 T | 0.1 T | 0.3 T | 0.1 T | 0.1 T |
| 24.6 | 40 | 106.90 | 0 | 1.98 | 4.02 | 100 | 98.15 | 96.24 |
| 28.7 | 60 | 381.63 | 2.49 | 33.91 | 51.82 | 99.35 | 91.11 | 86.42 |
| 32.5 | 80 | 799.70 | 21.49 | 135.07 | 181.70 | 97.35 | 83.11 | 77.28 |
| 48.5 | 100 | 1318.33 | 49.23 | 274.33 | 358.90 | 96.27 | 79.19 | 72.78 |
| 54.9 | 120 | 1648.33 | 71.37 | 372.10 | 484.77 | 95.67 | 77.43 | 70.59 |

In addition, the shielding performance of tungsten and lead plates of the same thickness was compared. The results are shown in Figure 6. The trend in the shielding performance of both plates is similar, indicating that a tungsten shielding plate can be used instead of lead. Furthermore, in the high-energy range, the 0.3 mm tungsten plate exhibited a better shielding performance than the lead plate.

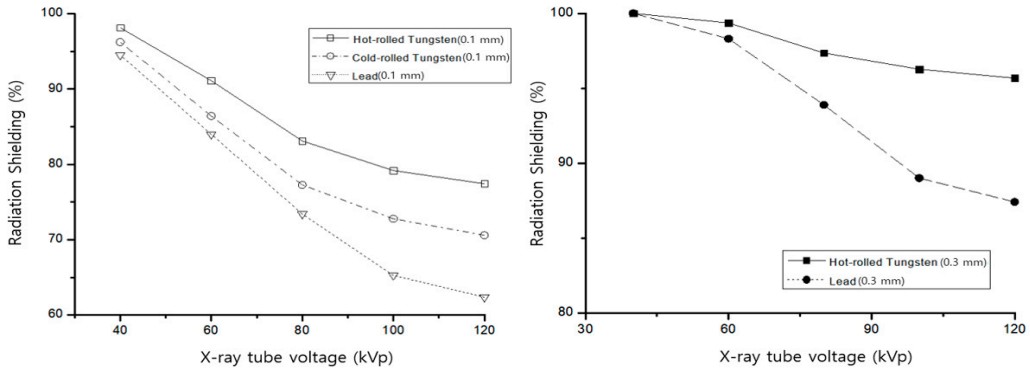

**Figure 6.** Comparison of shielding performance between the manufactured thin-film tungsten plate and lead.

## 4. Discussion

Most shielding plates used in medical institutions are used for shielding X-rays and gamma rays. Because the energy range is fixed, the plates used in medical institutions are observed to correspond to a smaller range than those used in working areas [33].

Generally, tungsten is mixed with an inorganic compound, such as resin, and processed into the desired shape and thickness. However, maintaining the reproducibility of the shielding performance is challenging. In this study, tungsten plates were used to fabricate shielding plates with different thicknesses and different processing methods. The three shielding plates produced had a thickness of 0.3 mm or less, and the difference in shielding performance between the plates produced via hot and cold rolling was less than 6%, indicating that there was no significant difference. However, it could have an impact on the materials used for small parts or on the production of custom shields.

In previous studies, the hot-rolling method was reported to reduce the thickness of the tungsten plate. Because this process could increase both durability and density, the shielding rate could also increase [34,35]. The shielding rate can be increased using a high-density shielding material or by increasing the density of the shield [36].

The hot and cold rolling methods are carried out at temperatures greater than and less than 1200 °C, respectively, and can be performed more effectively by preheating [37]. The cost of processing tungsten into a plate is high; however, it is eco-friendly and can be

used continuously. In addition, its excellent durability makes it a suitable device for use in medical institutions [38,39]. The thinner the tungsten plate, the more appropriate it is to use a cold-rolling method with a relatively simple processing technology. However, when the cross-sectional structures of the plates were evaluated, hot rolling was observed to be more appropriate for analyzing laminated structures. In particular, the same difference was observed in the X-ray area, which corresponds to the energy range of the diagnostic area.

When the shield is manufactured in the form of a sheet or film, voids may be generated in the process of stirring the mixture and shielding material. When the shielding plate is manufactured using the rolling method, the flexibility is lowered; however, the problem of such voids can be resolved. Moreover, tungsten plates can also help mitigate problems, such as hardening, and reduce cracks that occur in the long term [40,41].

Tungsten metal processing has limitations in terms of time and economic efficiency. However, the 0.3 mm-thick tungsten plate used in this study showed an excellent shielding performance as a replacement for lead; therefore, it can be considered as an eco-friendly material for use in radiation shielding in the future.

## 5. Conclusions

In this study, 0.3 mm and 0.1 mm pure tungsten plates were manufactured using hot and cold rolling processes to produce a shielding plate that could replace lead. In addition, the dependence of the shielding performance on the processing technology and thickness was evaluated and compared. Optical microscopy was performed to analyze the cross-section of the prepared shielding film. It was observed that the multilayer structure of the thin film was well formed in the hot rolling process. Moreover, in the case of hot rolling, the shielding performance at 100 kVp was observed to be approximately 80% for a thickness of 0.1 mm. Contrastingly, a shielding performance of 96% was observed for a thickness of 0.3 mm. Therefore, it was inferred that the shielding performance was superior to that of lead with the same thickness.

**Funding:** This work was supported by a National Research Foundation of Korea (NRF) grant funded by the Korean government (MEST) (NRF, 2020R1I1A3070451).

**Institutional Review Board Statement:** Not applicable.

**Informed Consent Statement:** Not applicable.

**Data Availability Statement:** Data are contained within the article.

**Conflicts of Interest:** The authors declare no conflict of interest.

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
