# Peer review of "Improving the X-ray Shielding Performance of Tungsten Thin-Film Plates Manufactured Using the Rolling Technology"

_applsci, doi:10.3390/app11199111_

Round 1
Reviewer 1 Report
The manuscript presents a useful study on the effect of thickness and manufacturing process of tungsten-based shielding plates on shielding performance. The obtained data can be useful for improving the radiation shielding performance of materials by improving manufacturing technology. However, several issues need to be addressed and clarified as below before the paper can be considered for publication.
- “Using a 1 mm plate (Beijing Tianlong tungsten Technology Co. Ltd), 0.3 mm and 0.1 mm plates were manufactured via hot rolling, and 0.1 mm plates were manufactured via cold rolling.” (Line 94) Please give the actual measured value of tungsten plate thickness and give the test error. This is critical to the authenticity of the shielding performance data.
- In general, tungsten is slowly oxidized under heating conditions, and this is confirmed by your Energy-dispersive X-ray spectroscopy. (Line 144) How does oxidized tungsten on the surface affect the shielding performance? Should a protective atmosphere (N2 or Ar) be used to exclude interference when processing tungsten at different temperatures?
- “Changes to the surface of the tungsten plate due to the different rolling methods were observed, as shown in Figure 5, and it was found that the surface was slightly denser in the 0.1mm plate repeatedly forged to reduce the thickness.” Please provide more detailed density data for samples of different thicknesses and different preparation processes in the manuscript. (Line 156)
In a summary, this work is well-organized, and I suggest that the manuscript can be published after minor revision.
Reviewer 2 Report
Dear Author,
Please follow my comments in the attached file.

Round 2
Reviewer 2 Report
Dear Author,
Thank you very much for your extensive work on the revision of your manuscript. Your answers satisfied me and I found some interesting information in your answers, so I thank you for letting me know something new. Now I agree to accept it for publication.
Best regards,
Reviewer